# The Effects of Environmental Regulation and Low-Carbon Logistics Capacity on the Level of New Urbanization in Six Central Provinces of China

**Yifan Wang \*** , **Zhongfu Yu** and **Yamin Hou**

College of Economics and Management, Xidian University, Xi'an 710126, China
\* Correspondence: wyf635803216@163.com

**Abstract:** The urbanization of a region is affected by the implementation of various policies, and to explore the specifics of the environmental regulation at today's new level of urbanization, the increased logistics capacity of a region and the consequent carbon emissions must be the focus of our attention. For the values considered by the study, the six central provinces of China have obvious location advantages and urban–rural differences, so a static panel regression effect model was constructed based on the inter-provincial panel data of the six central provinces of China from 2005–2019, and the entropy weight method was applied to quantify the low-carbon logistics capacity and new urbanization level in the region. The model explores the relationship between environmental regulation, regional low-carbon logistics capabilities, and the level of new urbanization. The results of the study show that the levels of new urbanization in all six provinces are increasing rapidly, year on year. Environmental regulation has a positive impact on regional low-carbon logistics capabilities and the level of new urbanization, and environmental regulation promotes the improvement of the level of new urbanization through a significant positive impact on regional low-carbon logistics capabilities, and there is an intermediary conduction effect. This paper provides valuable reference suggestions for low carbon development and new urbanization in six central provinces through empirical research.

**Keywords:** environmental regulation; low-carbon logistics capabilities; new urbanization levels; intermediary effect

## 1. Introduction

Cities are now the center of much human activity, and the growing desire for a better quality of life and access to better education and health services has led to mass migration from rural to urban areas. Chinese urbanization has never stopped moving forward. We often measure the urbanization rate as the share of the urban population in the total population, and the Chinese urbanization rate, which was 17.92% in 1978, has grown exponentially to 60.6% by 2019. However, compared to the overall urbanization level of more than 75% in developed countries, the Chinese urbanization level was still at a relatively low level. According to the United Nations Development Organization, China is expected to have 70% of its urban population and 1 billion people living in cities and towns by 2030. Chinese urbanization process will continue to grow at a high rate for a considerable period.

However, Chinese rapid urbanization will generate huge energy demand. Energy consumption per capita in urban areas is almost six times higher than in rural areas, which means that urbanization will put enormous pressure on energy. As the second largest economy in the world today, China needs to balance rapid economic development with environmental protection, and urbanization should not come at the cost of intensive energy consumption and environmental pollution. In March 2014, the Central Committee of the Communist Party of China (CPC) and the State Council jointly released the National

New-Type Urbanization Plan (2014–2020), which aimed to promote energy conservation through urbanization. Large-scale infrastructure projects in the city would consume a lot of steel, cement, and other raw materials, and the production of raw materials would also consume a lot of energy and, therefore, would produce a large amount of pollutants, which would cause serious environmental pollution problems. The rising standard of living and the increasing consumer demands of urban residents would not only increase the consumption of energy-intensive goods but also promote the demand for logistics and distribution. Urban dwellers consume more energy than rural dwellers and therefore produce a lot of greenhouse gases. Therefore, it is necessary to explore the relationship between urban development and environmental regulation.

Logistics, as a comprehensive service industry integrating transportation, warehousing, freight transportation, information, and other comprehensive services, is the main pillar industry of China's national economic development. Investment in the logistics sector increase economic efficiency, reduce transaction costs, and increase resource utilization and productivity. Economies with better logistics networks have higher economic returns [1]. However, logistics operations also contribute significantly to greenhouse gas (GHG) emissions. As reported in this regard, the World Economic Forum highlighted that logistics emissions accounted for 5.5% of global emissions [2]. Inefficient logistics pose many environmental challenges, including excessive GHG emissions, noise, waste, and high fuel combustion. The country has begun to focus on how to use policy instruments to promote the development of a green and low-carbon economy, to base development on the efficient use of resources and the effective control of greenhouse gas emissions, to establish a green, low-carbon, and circular development economic system, and to ensure that the goal of "peak carbon emissions and carbon neutrality" is achieved. To this end, China also advocates the establishment of a low-carbon transportation system, accelerating the development of low-carbon transportation modes such as railroad transportation and water transportation, and popularizing low-carbon transportation modes such as aviation and navigation, thus being able to comprehensively promote the development of low-carbon logistics.

From the current state of the national economy, regional balanced development had been proposed for many years, but the gap between the east and the west objectively existed, and it was not a matter of trying to narrow the gap overnight. To solve the problem of regional development disparities in China, the six central provinces play an important role in the geographical division of labor in China. These provinces in the central region are basically along the Beijing–Guangzhou and Beijing–Kowloon railroads, and most of them are along both the Yangtze and Yellow rivers or the Eurasian Continental Bridge. They have a special role in geographically running north–south and taking over east–west. Taken together, the central region has high research value in terms of ecological environment construction, logistics and transportation collection level as well as new urbanization development.

Above all, due to the effectiveness of environmental regulation, the linkage between low-carbon development and logistics development, and the importance of urbanization for economic development, the following issues have attracted a lot of attention in China. What factors will affect the level of new urbanization in a region? Does the intensity of environmental regulation have an impact on the level of new urbanization through some kinds of mediated transmissions? Will a more environmentally friendly logistics industry have a synergistic effect on the development of Chinese urbanization? What is the relationship between environmental regulation, regional low-carbon logistics development capacity, and new urbanization level? The answers to these questions have important implications for urbanization policy and environmental governance. There are relatively few studies on the impact of environmental regulation and regional low-carbon logistics capacity on new urbanization in the homeland and abroad. This research considered the impact of other factors on urbanization more than environmental factors [3–7]. To address these issues, this study will use the most representative six provinces in central

China (Jiangxi, Anhui, Hubei, Shanxi, Hunan, and Henan) as the sample, focusing on the entropy power method to measure the level of new urbanization and regional low-carbon logistics capacity and to study the impacts of environmental regulation on both of them. The interrelationship between environmental regulation, regional low-carbon logistics capacity, and new urbanization level is studied to provide a policy reference basis for the development of low-carbon logistics and new urbanization in the central region.

The main contributions of this research are as follows:

(1) Based on the application of the mediating effect panel model, this paper revealed that the effect of environmental regulation on the level of new urbanization was indirect, and regional low-carbon logistics capacity as a special variable assumed a local mediating transmission role.

(2) This paper used a fixed-effects panel regression model combined with a mediated-effects model, which enabled a comprehensive analysis of the extent to which new urbanization development varied across provinces in different periods in both time and regional dimensions.

(3) In the long run, the rapid development of the logistics industry will bring a dramatic increase in carbon emissions, thus putting pressure on environmental protection. Therefore, the low-carbon transformation of the logistics industry is particularly important. Considering that the low-carbon logistics capacity of a region is related to various factors, this paper proposed a new concept of regional low-carbon logistics capacity and combined multidimensional factors to quantify the low-carbon logistics capacity of a region by using the method of index construction. These works provide a unique perspective for the study of measuring a region's environmental capacity and provide valuable references for China to better achieve sustainable development.

The remainder of this paper is organized as follows. In Section 2, we provide a comprehensive literature review of research on environmental regulation, regional low-carbon logistics capacity, new urbanization level, and other related aspects. Next, in Section 3, we explain how to construct an indicator system for the new urbanization level of six provinces in central China, and select the entropy method from various indicator measurement methods to measure it. In Section 4, we describe the analytical approach used in this study to assess the direct effects of environmental regulation on the level of new urbanization versus those transmitted through intermediaries and develop the corresponding regression models, followed by detailed explanatory notes on the various variables selected. The empirical study and results of the model are discussed in Section 5. Finally, in Section 6, we present a discussion that analyzes the conclusions of this study and the reasons for such results and suggest corresponding policy recommendations.

## 2. Literature Review

The research in this paper involved three main modules, which were environmental regulation, regional low-carbon logistics capacity, and new urbanization level. First, studies of the impacts of environmental regulation on various developments in a region have been subject to varying academic views. The influential "Porter hypothesis" suggested that appropriate environmental regulation helped "push" firms to innovate green technologies, resulting in "compensating benefits" that exceeded the costs of environmental regulation. So, proper environmental regulation was going to promote the development of a region [8]. On this basis, companies could apply the results of green technology innovations to subsequent production processes, moving them away from the original polluting production model and reducing the additional costs caused by environmental controls [9]. Although the binding nature of environmental regulation might initially inhibit the economic efficiency of businesses, it promotes business development in the long run [10]. Emissions trading policies, as one of the main instruments of environmental regulation, play an important role in the development of businesses in a region. However, current studies on the effects of emissions trading policies mainly focused on innovation performance in the context of developed countries, and it was found that the implementation of

emissions trading policies could significantly improve the green innovation performance of a region, but also inhibit the green innovation capacity of neighboring regions at the same time [11]. Other policy instruments, such as the implementation of energy policies, could also have an impact on businesses. Increased urban liberalization might enhance the contribution of conventional energy policies [12]. On this basis, we found that the impact of the policy was closely related to the development of the city. Just as the level of green technology innovation was also highly correlated with the level of urbanization, green technology innovation was the dominant force [13]. Urbanization could also promote the aggregation of urban population and economic activities, which in turn increased the total amount of innovation activities. Meanwhile, the scale of science and technology innovation activities was significantly and positively correlated with the level of urbanization [14]. So, in summary, it was easy to see that environmental regulation had an impact on the urbanization of a region. The phrase "new type of urbanization" had entered the public eye because the national government had integrated the concept and principles of ecological civilization into the whole process of urbanization, and urbanization had received wide attention from the academic community [15] and it was not difficult to see that the focus of the new-type urbanization was also ecological construction. The new-type urbanization strategy proposed by the Chinese government was human-centered urbanization, which emphasized the coordination of population, economy, society, and ecological environment. Moreover, the new-type urbanization had a significant energy-saving effect, with the effect being greater in resource-rich areas [16]. The implementation of environmental regulation could also strengthen the role of new urbanization in promoting regional growth and optimizing the rational allocation of resources. In addition, the spillover effect of environmental regulation on the level of new urbanization reflected spatial heterogeneity and promoted the level of new urbanization in the surrounding areas [17]. However, it was not clear in what way environmental regulation affected the level of new urbanization.

When discussing the topic of environmental protection and development, energy consumption and carbon emissions are inevitably highly involved. According to a 2009 report by the World Economic Forum, logistics and transport activities contributed about 2800 megatons of $CO_2$, which was equivalent to 5.5% of the annual $CO_2$ emissions generated by human activities. If the development of low-carbon logistics could reduce more than 50% of pollution emissions, about 1400 megatons, the common ways were clean vehicle technology, optimization of transportation networks, etc. [18]. In China, the logistics industry was included in the government's ten major industrial adjustment and development plans, which indicated that the logistics industry was an important part of the national economy and would receive policy and financial support and that the impact of environmental regulation on industrial control would also be significant for the logistics industry. The logistics industry was an important industry that connected production and consumption, and it involved transportation, warehousing, distribution, and many other links, in each of which carbon emissions were generated. Therefore, this required the integration of low-carbon concepts in the process of logistics system improvement to protect the environment while developing the economy [19]. Currently, many scholars have conducted specific studies on regional low-carbon logistics. There were different approaches to measuring low-carbon logistics capabilities. M Lu, R Xie, et al. constructed an Environmental Logistics Performance Index (ELPI) to assess the overall performance of 112 countries in green transport and logistics practices. The ELPI effectively represented the trade-off between logistics efficiency and environmental protection in transport [20]. He Z, Chen P, Liu H, et al. developed a generic PMS containing 42 indicators in 12 dimensions for assessing low-carbon logistics based on a triple bottom line framework using a multicase study and literature analysis. It also provided suggestions for the construction of a low-carbon logistics system in China from six aspects [21]. Yang L, Cai Y, Zhong X, et al. instead focused on studying port logistics, which is an important node of logistics and whose energy consumption accounts for a considerable proportion of the transportation industry. Taking Shenzhen port as an example, they proposed a method to measure the

carbon emissions of the port integrated logistics system [22]. Based on the analysis of the structural characteristics and dynamic feedback of the low-carbon logistics system, Yang Guanghua used the basic principles and methods of system dynamics to establish a dynamics model of the low-carbon logistics system [23]. Leaving aside the study of domestic regions, if we looked at oil-producing countries in sub-Saharan Africa, we would find that in addition to the interaction of consumption capacity and income level with carbon emission intensity, the increase in urbanization level also increased the intensity of carbon emissions to a certain extent [24]. However, most of the current research analysis on low-carbon logistics focused on a single dimension such as logistics performance and carbon emission intensity [25,26]. There were few findings on the comprehensive measurement of low-carbon logistics in a region and the impact of low-carbon logistics capabilities derived in the context of environmental regulation on other factors in a region.

Nowadays, the rapid progress of the logistics industry has a huge pulling effect on the development of regional urbanization, and there is a significant linkage between the development of the logistics industry and the growth of the urbanization rate [27]. In the early academic world, Adam Smith pointed out in *An Inquiry into the Nature and Causes of the Wealth* that due to the constraints of logistics, the initial space for economic activities was always distributed in places with easy access [28]. Cities and towns tend to be more developed than villages in terms of transportation construction. The famous economist Chinnery also said that the promotion of urbanization has an optimization effect on the primary industry, an enhancement effect on the secondary industry, and a significant promotion effect on the logistics industry. From another point of view, the reasonable adjustment and optimization of the logistics industry needed to be considered in line with the trend of urbanization and to match the scale and development of the town. Taking the eastern, central, and western regions of China as the research objects, Qian L analyzed the correlation effect between the development of the logistics industry and the urbanization rate, and the results of the study found that there is a strong connection between them [29]. In terms of measuring carbon emissions, similar studies had been conducted to estimate the total carbon emissions of the logistics industry in China by provinces and cities using the IPCC carbon accounting method and to measure and decompose the regional differences in carbon emissions of the logistics industry in China by introducing the Theil index and regional separation coefficients [30]. In conclusion, it could be concluded that the development of the logistics industry was closely related to the new urbanization, but there was no definite conclusion on the impact of low-carbon logistics capacity on urbanization, which was something we needed to explore further.

Throughout the previous literature [31–34] descriptions, domestic and foreign scholars had conducted a lot of research on environmental regulation, regional low-carbon logistics capacities, and new urbanization levels, but there were still no explored aspects. First, previous studies had focused on the role of environmental regulation on green technology innovation or high-quality economic development in a region, but few had studied the impact of environmental regulation on urbanization in a region, and even fewer had introduced the concept of new urbanization [35,36]. Second, the current academic research on regional low-carbon logistics capacity was mostly limited to the measurement and evaluation aspects [37,38]. These studies did not explore whether regional low-carbon logistics capacity would have a profound impact on the development of a region or whether it would act as a bridge between environmental regulation and the new urbanization development. Then, most existing studies focused on the impact of the interaction between environmental regulation and new urbanization level on regional economic development, and less on the impact of other factors on the new urbanization level as the evaluated subject [39,40]. Finally, with the regionalization of the Chinese economy, the Yangtze River Delta, the Pearl River Delta, the Northeast Economic Zone, the Bohai Sea Rim, and the Chengdu-Chongqing Economic Zone had all become the subjects of keen research by domestic scholars due to their rapid economic development. We often overlook the value of the six central provinces as transportation hubs and population concentrations, which

have more research value in environmental protection construction, the logistics industry, and urbanization development. Therefore, this paper used econometric analysis to explore the relationship between environmental regulation, regional low-carbon logistics capacity, and new urbanization levels using the panel data of six central provinces from 2005 to 2019 and established a static fixed-effects panel model and a mediated transmission model to combine theoretical and empirical analyses to provide innovative reference suggestions for the urbanization development of six central provinces.

## 3. Selection and Measurement of Indicators

### 3.1. Indicator Selection

Today's rapid urbanization process inevitably brings problems in terms of uncoordinated development, slow economic transformation, and environmental pollution. Therefore, nowadays, with the emphasis on environmental regulation, it is necessary to follow a new green and efficient path of urbanization development. The new urbanization is also a people-centered urbanization process, and the satisfaction of urban residents plays a very important role in the good development of the city. In terms of urban population growth, it is important to protect the interests of the population moving into the city, to absorb more people moving to the city in terms of production and living, and to truly realize the transformation of urbanization. In terms of economic development, we should focus on the agglomeration and upgrading of industries in the town, improve the radiation level of new industries, and drive the high-quality development of the region. In terms of the ecological environment, it is necessary to create a beautiful and livable urban environment, pay attention to the construction of ecological civilization, control the pollution emission intensity of relevant enterprises, and improve the green level of the region from the source. In terms of urban–rural coordination, the key role of new urban–rural integration is reflected in aspects such as the reduction of the income consumption gap between urban and rural residents, and the elimination of the differences between the urban and rural dichotomy in China is one of the important goals of new urbanization. Finally, only the improvement of urban infrastructure can provide the necessary support for the development of new urbanization and make the beauty of urban life visible in concrete terms.

In this paper, the index construction of the new urbanization level was based on the relevant research literature of previous scholars and combined with the requirements of scientificity, objectivity, systematization, and data availability [41,42]. The index system is constructed by selecting indicators from five dimensions of the new urbanization level: population, economy, ecology, urban–rural, and infrastructure, and its details are shown in Table 1.

### 3.2. Indicator Measurements

The research object of this paper was the six provinces in central China (Jiangxi, Anhui, Hubei, Shanxi, Hunan, and Henan) and the selected study period was 2005–2019. The entropy weight method was chosen to measure the new urbanization level of each province in terms of population, economy, ecology, urban and rural areas, and infrastructure. There are many methods to measure the index system, such as AHP, factor analysis, and principal component analysis, but these methods are more subjective in measuring the index weights and lack the accuracy of the study. While the entropy value of each index in the entropy method is the disorder of information, in general, the smaller the entropy weight the lower the degree of disorder of the system, and the closer the information of the index is indicated. The level of new urbanization is a variable reflecting objective facts, which requires concrete and objective data for analysis. Therefore, using the entropy weighting method to determine the weights can minimize the influence of human factors on the weights, thus improving the credibility and scientificity of the assessment.

The detailed calculation steps of the entropy weight method are as follows: (1) First, normalize $X_{ij}$; (2) Then calculate the entropy value of the $j$th indicator $E_j$, Normally let

$0 \leq E_j \leq 1$, generally take $k = 1/\mathrm{n}\ m$; (3) Finally, the weight coefficient $W_j$ of information quantity is calculated.

$$p(X_{ij}) = \frac{X_{ij}}{\sum_{i=1}^{m} X_{ij}} (i = 1, 2, \ldots, m; j = 1, 2, \ldots, n)$$
$$E_j = -k\sum_{i=1}^{m} p(X_{ij}) \ln p(X_{ij})$$

$$d_j = 1 - E_j$$

$$W_j = \frac{d_j}{\sum_{i=1}^{m} d_j} (j = 1, 2, \ldots, n)$$

In each of the above formulas, $p(X_{ij})$ is the standardized value of each indicator, $E_j$ is the information entropy value, $W_j$ is the weight of each indicator.

**Table 1.** New urbanization level indicator system.

| System Layer | Subsystem Layer | Indicator Layer | Indicator Positivity (+) and Negativity (−) |
|---|---|---|---|
| New urbanization level | Population | Urbanization rate (%) | + |
| | | Urban population density (people/square kilometer) | + |
| | Economy | GDP per capita (CNY) | + |
| | | Per capita disposable income of urban residents (CNY) | + |
| | | Share of non-agricultural industries in GDP (%) | + |
| | Ecology | Greening coverage of built-up areas (%) | + |
| | | Urban green space per capita (square meters) | + |
| | | Industrial wastewater discharge (10,000 tons) | − |
| | | Industrial waste gas emissions (tons) | − |
| | | Industrial solid waste generation (10,000 tons) | − |
| | Urban and rural | Ratio of per capita disposable income of urban residents to per capita disposable income of rural residents (Comparison based on rural residents = 1) | + |
| | | Ratio of per capita consumption expenditure of urban residents to per capita consumption expenditure of rural residents (Comparison based on rural residents = 1) | + |
| | Infrastructure development | Road area per capita (square meters) | + |
| | | Urban water penetration rate (%) | + |
| | | City gas penetration rate (%) | + |
| | | The proportion of built-up area to urban area (%) | + |

## 4. Model Construction, Variable Selection and Data Description

### 4.1. Model Construction

4.1.1. A Static Panel Fixed Effects Regression Model of Environmental Regulation on the Level of New Urbanization

First, to empirically test the above research hypotheses, the following basic model was constructed for the direct transmission mechanism of environmental regulation on the level of new urbanization. This model can very intuitively reflect the trends in different regions and thus analyze the interactions that exist between various variables.

$$NUR_{\mathrm{it}} = \lambda + \alpha_1 ER_{it} + \sum \alpha_j X_{it} + \varepsilon_{it} \tag{1}$$

In the above equation, the explanatory variable $NUR_{\mathrm{it}}$ denotes the level of new urbanization in province i at year t, $ER_{it}$ denotes the level of environmental regulation in province i in year t, $X_{it}$ denotes a set of control variables that affect the level of new urbanization; $\varepsilon_{it}$ is the random perturbation term. $\lambda$ measures the dynamic change of $NUR_{\mathrm{it}}$; $\alpha_1$ measures the total effect of $ER_{it}$ on $NUR_{\mathrm{it}}$.

4.1.2. Regression Model of the Mediating Effect of Environmental Regulation on the Level of New Urbanization

The above study reveals the transmission relationship between regional low-carbon logistics capacity and environmental regulation and new urbanization. To corroborate the hypothesis of the study, the following recursive type model of the mediating mechanism is constructed. In the first step, the direct effect of environmental regulation on the level of new urbanization was tested, as shown in Equation (1). In the second step, based on the previous theoretical analysis, regional low-carbon logistics capacity (LCLC) was set as the mediating variable. To test the significance of the regression of the explanatory variable environmental regulation on regional low-carbon logistics capacity and the significance of the regression of the mediating variable regional low-carbon logistics capacity on the explanatory variable new urbanization level, the following model was constructed:

$$LCLC_{it} = \lambda + \beta_1 ER_{it} + \sum \beta_j X_{it} + \varepsilon_{it} \tag{2}$$

$$NUR_{it} = \lambda + \beta_2 LCLC_{it} + \sum \beta_j X_{it} + \varepsilon_{it} \tag{3}$$

In the above equations: the intermediate variable $LCLC_{it}$ denotes the regional low-carbon logistics capacity of province i in year t, $\beta_1$ and $\beta_2$ measure the total effect of $ER_{it}$ on $LCLC_{it}$ and the total effect of $LCLC_{it}$ on $NUR_{it}$, respectively.

If both $\beta_1$ and $\beta_2$ pass the significance test, the regional low-carbon logistics capacity (LCLC) is included in the final mediated effects estimation model, construct the following model:

$$NUR_{it} = \lambda + \varphi_1 ER_{it} + \varphi_2 LCLC_{it} + \sum \varphi_j X_{it} + \varepsilon_{it} \tag{4}$$

In the above equation: $\varphi_1$ and $\varphi_2$ measure the total effect of $ER_{it}$ on $NUR_{it}$ and the total effect of $LCLC_{it}$ on $NUR_{it}$, respectively. If they both pass the significance test, it indicates the existence of a mediating mechanism between environmental regulation and the level of new urbanization. In terms of effect size, the direct effect of $ER_{it}$ on $NUR_{it}$ accounts for $\varphi_1/\alpha_1$, and the mediating effect of regional low-carbon logistics capacity accounts for $\varphi_2\beta_1/\alpha_1$.

*4.2. Variable Selection*

4.2.1. Dependent Variable

The new urbanization level (NUR) is the dependent variable. Since the connotation of new urbanization involves a wide range, it is easy to deviate by using only one indicator to express it. Accordingly, this paper constructed a new urbanization evaluation index system from five dimensions: population, economy, ecology, urban-rural, and infrastructure, and used the entropy weight method to measure the new urbanization level of each province specifically. The details of the indicator system are shown in Table 1.

4.2.2. Independent Variable

Environmental regulation (ER) is the core independent variable. Previous literature has varied in its approach to measuring environmental regulation. Regulation is a regulatory and policy constraint set by the government for economic activities, so environmental regulation is, by definition, direct or indirect control and intervention by the government on the use of resources and business activities of enterprises. The government can rely on this approach to finally go green. Since Chinese current environmental management is also dominated by administrative means, this paper used the proportion of investment in environmental pollution control to GDP to express the intensity of environmental regulation. If the proportion of investment in environmental pollution control to GDP is larger in a region, it means that the intensity of environmental regulation in that region is also larger. The data on the share of investment in environmental pollution control in GDP for each

province in this study were obtained from the China Environmental Statistical Yearbook of previous years.

### 4.2.3. Mediated Variable

The regional low-carbon logistics capacity (LCLC) is the mediating conductive variable in this study. Logistics is an important part of the national economy and involves a wide range of fields. Logistics plays an important role in promoting industrial restructuring, transforming the model of economic development, and enhancing the competitiveness of the national economy. With the rapid development of regional logistics capacity, there are bound to be a lot of waste emissions, air pollution, energy consumption, and other problems, especially the problem of increasing greenhouse gas emissions such as carbon dioxide in transportation. In response to the above issues and concerning previous scholars' research, this paper defined regional low-carbon logistics capacity: the capacity of a logistics supply body that aims at high energy efficiency and low emissions can provide the required logistics services to logistics demand bodies in each region by effectively organizing and utilizing the internal resources of its logistics system within a specific period. Therefore, the regional low-carbon logistics capacity can visually and quantitatively reflect the level of logistics development and ecological construction of a region and is a comprehensive variable.

According to the definition, the regional low-carbon logistics capacity itself is a complex structure, with many layers, each system will exist between the interaction of a whole, so in the construction of the index system, the combination of qualitative and quantitative analysis means can be a more scientific evaluation of regional low-carbon logistics capacity.

In the process of evaluation, the search for indicators that best represent the regional low-carbon logistics capacity should be studied as a complex system. First, to fully reflect the essential characteristics of regional low-carbon logistics capacity, we decomposed it into four basic elements: infrastructure capacity, environmental protection capacity, business development potential and the low-carbon ecological level which was the focus of previous scholars' studies. Next, after the basic elements were selected, we needed to further decompose them into different indicators to constitute the whole system, as shown in Table 2.

### 4.2.4. Controlled Variables

The level of green innovation (GTI) is one of the most important factors affecting the frontier development of a region. Existing studies do not yet have a uniform metric for green innovation. There are two main methods of measurement. The first approach is to measure the level of green innovation in terms of the number of patents for simple technological inventions. However, many patents on technological inventions are not used in actual production and do not have a real impact on green innovation, and thus cannot be used as a measure. The second approach is to use the variables "green product innovation" and "green technology innovation" to measure "green innovation". For the accuracy of the study, this paper improved the measurement method and adopted the entropy weight method to measure the level of green innovation, as shown in Table 3 for the specific index system.

Industrial structural upgrading (ISU) can have a significant impact on the size of a region's economy and urbanization development. This paper used the ratio of the value-added of the tertiary sector to the value-added of the secondary sector in each province to measure the level of industrial structure upgrading.

Foreign direct investment (FDI) can bring new development space to a region, and it will affect the new urbanization in the economic aspect. This paper used the ratio of the actual amount of foreign direct investment utilized in each province to the GDP of that province to measure the strength of foreign investment in it.

**Table 2.** Regional low-carbon logistics capacity indicator system.

| System Layer | Subsystem Layer | Indicator Layer | Indicator Positivity (+) and Negativity (−) |
|---|---|---|---|
| Regional low-carbon logistics capacity | Infrastructure capacity | Total postal line length per capita (kilometers per 10,000 people) | + |
| | | Freight volume per capita (tons/person) | + |
| | | Cargo turnover per capita (one hundred million-ton kilometers per 10,000 people) | + |
| | | The business volume of express delivery per capita above the scale (Number/person) | + |
| | | Logistics GDP per capita (CNY 100 million per 10,000 people) | + |
| | | Logistics GDP as a share of total provincial GDP (%) | + |
| | | Urban road area per capita (square meters) | + |
| | Environmental protection capacity | Total postal and telecommunications services per capita (CNY 100 million per 10,000 people) | + |
| | | Total import and export of goods per capita (USD/person) | + |
| | | Cell phone penetration rate (%) | + |
| | | Long distance fiber optic cable line length (kilometers) | + |
| | Business development capability | Share of education spending in fiscal spending (%) | + |
| | | Logistics industry urban new fixed asset investment (CNY 100 million) | + |
| | | The share of logistics workers in the total population of the province (%) | + |
| | | Average number of students enrolled in higher education institutions per 100,000 population (people) | + |
| | | Expenditure on research and experimental development (CNY 100 million) | + |
| | | Financial industry value added (CNY 100 million) | + |
| | Low-carbon ecological level | Carbon emission intensity of logistics industry (1000 tons of $CO_2$/CNY 100 million) | − |
| | | Forest coverage (%) | + |
| | | Total energy consumption as a percentage of total provincial GDP (%) | − |

**Table 3.** Green innovation level indicator system.

| System Layer | Subsystem Layer | Indicator Layer | Indicator Positivity and Negativity |
|---|---|---|---|
| The level of green innovation | Green technology innovation | Number of patents granted (items) | + |
| | Green product innovation | Product sales revenue/total energy consumption (CNY 1000/10,000 tons of standard coal) | + |
| | Green process innovation | Internal expenditure on R&D expenses (CNY thousand) | + |

Fixed capital (GFCF) is one of the key factors to improve the productivity and economic development of a region. This study used the ratio of the total fixed capital formation of each province to the average of the year-end resident population of that province to measure the level of fixed capital in that province.

The basic demographic profile (BDS) can greatly influence the urbanization of a region. The aging problem of the Chinese society is serious at present, so this paper adopted the population aging situation to represent the basic situation of the population. This was carried out by using the ratio of the number of people sampled aged 65 years and older

to the number of people sampled aged 15–64 years in each province to measure the basic population profile of that province.

### 4.3. Data Description

The data in this paper were obtained from China Statistical Yearbook, China Energy Statistical Yearbook, China Environmental Statistical Yearbook, Wind database, and other statistical databases. In the process of data processing, the following principles are followed: (1) Data availability. Since there are no systematic and specialized statistics on the logistics industry in various statistical departments in China, this paper adopted the data related to the "transportation, storage and postal industry" of six central provinces from 2005 to 2019 to measure the regional low-carbon logistics capabilities, drawing on the research of previous scholars capacity. (2) Accuracy of data. For example, in the measurement of new urbanization level and regional low-carbon logistics capacity, many previous papers considered the original value of the data provided in the statistical yearbook as the index value for calculation, but in this paper, after considering the development differences of each region, data such as freight volume and total import and export of goods were processed per capita, so that the index data could be measured more accurately. (3) Completeness of data. The exponential smoothing method was used to make up for the missing data in individual years. The explanatory notes on the variables are shown in Table 4. The results of descriptive statistics of the variables are shown in Table 5.

**Table 4.** Explanatory notes for each variable.

| Variable Name | Variable Symbol | Explanatory Notes | Unit |
|:---:|:---:|:---:|:---:|
| New urbanization level | NUR | Entropy values for the six provinces | - |
| Environmental regulation | ER | Total investment in industrial pollution control/total GDP | CNY 10,000/CNY 100 million |
| Regional low-carbon logistics capacity | LCLC | Entropy values for the six provinces | - |
| The level of green innovation | GTI | Entropy values for the six provinces | - |
| Advanced industrialization | ISU | Tertiary industry value added/secondary industry value added | - |
| Foreign investment | FDI | Actual amount of FDI utilized/total GDP | 10,000 USD/ CNY 100 million |
| Fixed capital | GFCF | Total fixed capital formation/number of resident population at the end of the year | CNY 100 million/10,000 people |
| Population basic information | BDS | Number of sampled population aged 65 and above/number of sample population aged 15-64 | - |

**Table 5.** Descriptive statistics of variables.

| Variable Name | Mean | Std.DEA. | Max | Min | Number of Observations |
|---|---|---|---|---|---|
| NUR | 0.4841975 | 0.095679 | 0.6805449 | 0.2610147 | 90 |
| ER | 0.013315 | 0.0065966 | 0.0402823 | 0.0056233 | 90 |
| LCLC | 0.3117745 | 0.1662666 | 0.7689699 | 0.0792948 | 90 |
| GTI | 0.2548255 | 0.2271149 | 0.8892064 | 0.0069619 | 90 |
| ISU | 0.8506047 | 0.2238402 | 1.438985 | 0.4970531 | 90 |
| FDI | 35.47081 | 14.86263 | 61.89075 | 6.50415 | 90 |
| GFCF | 1.744317 | 0.8851637 | 3.486875 | 0.361781 | 90 |
| BDS | 0.1391829 | 0.0238002 | 0.2072887 | 0.0957742 | 90 |

## 5. Analysis of Empirical Results

### 5.1. Smoothness Test of Variables

To ensure that the empirical results are true and reliable, unit root LLC tests were conducted on the variables to ensure the smoothness of the data and prevent spurious regressions, and the test results are shown in Table 6. From the test results, we could see that the unit root test of the dependent variable NUR was not smooth, but its first-order difference variable D_NUR was smooth, and the unit root tests of other variables were all smooth and belong to a smooth series so that the cointegration test could be performed to avoid the pseudo-regression phenomenon.

**Table 6.** Variable stability test.

| Variables | t-Statistic | p-Value | Test Results |
|---|---|---|---|
| NUR | −1.1259 | 0.1301 | Unstable |
| D_NUR | −4.5950 | 0.0000 | Stable |
| ER | −3.0613 | 0.0011 | Stable |
| LCLC | −2.0421 | 0.0206 | Stable |
| GTI | −3.1316 | 0.0009 | Stable |
| ISU | −2.3657 | 0.0090 | Stable |
| FDI | −2.3429 | 0.0096 | Stable |
| GFCF | −3.0224 | 0.0013 | Stable |
| BDS | −2.7050 | 0.0034 | Stable |

### 5.2. Variable Co-Integration Test

For panel data with non-stationary cases, a cointegration test was performed before carrying out regression analysis to analyze whether there was a direct long-run equilibrium relationship between the variables. In this paper, the Kao cointegration test as well as the Pedroni cointegration test was conducted on the variables, and the results are shown in Tables 7 and 8. From the test results, it could be seen that both the Kao cointegration test and Pedroni cointegration test passed the significance test at a 1% significance level for each set of test statistics, so the original hypothesis was rejected and there was a long-term cointegration relationship between the variables so that regression analysis could be performed.

**Table 7.** Kao cointegration test.

| Kao Test for cointegration | | |
|---|---|---|
| **Ho: No cointegration** | Number of panels = 6 | |
| **Ha: All panels are cointegrated** | Number of periods = 13 | |
| **Cointegrating vector: Same** | | |
| **Panel means: Included** | Kernel: Bartlett | |
| **Time trend: Not included** | Lags: 1.33 (Newey-West) | |
| **AR parameter: Same** | Augmented lags: 1 | |
| | **Statistic** | ***p*-value** |
| **Modified Dickey–Fuller t** | −4.0471 | 0.0000 |
| **Dickey–Fuller t** | −4.5573 | 0.0000 |
| **Augmented Dickey–Fuller t** | −2.3758 | 0.0088 |
| **Unadjusted modified Dickey–Fuller t** | −4.0051 | 0.0000 |
| **Unadjusted Dickey–Fuller t** | −4.5484 | 0.0000 |

**Table 8.** Pedroni cointegration test.

| Pedroni test for cointegration | | |
|---|---|---|
| **Ho: No cointegration** | Number of panels = 6 | |
| **Ha: All panels are cointegrated** | Number of periods = 14 | |
| **Cointegrating vector: Panel specific** | | |
| **Panel means: Included** | Kernel: Bartlett | |
| **Time trend: Included** | Lags: 2.00 (Newey–West) | |
| **AR parameter: Panel specific** | Augmented lags: 1 | |
| **Cross-sectional means removed** | | |
| | **Statistic** | ***p*-value** |
| **Modified Phillips–Perron t** | 3.7344 | 0.0001 |
| **Phillips–Perron t** | –11.3705 | 0.0000 |
| **Augmented Dickey–Fuller t** | –9.0607 | 0.0000 |

*5.3. An Empirical Analysis of the Benchmark Regression Model of the Impact of Environmental Regulation on the Level of New Urbanization*

According to the results of the Hausman test, the chi-square value was 137.97 and the *p*-value was 0.000, which passed the 1% significance test and the original hypothesis should be rejected, so the individual fixed effects model of the static panel was selected for analysis in this paper. Table 9 displays the test results of the mixed regression model OLS, the individual fixed effects model FE, and the random-effects model RE in comparison. From the analysis results, although the mixed regression model and the random-effects model outperformed the individual fixed effects model in terms of overall variable significance, the individual fixed effects had a better fit in terms of the overall model goodness-of-fit R2. Therefore, combined with the results of the Hausman test, it was more appropriate to select the individual fixed-effects model for the analysis of the results.

**Table 9.** Empirical analysis of baseline regression model.

| Variables | OLS | FE | RE |
|---|---|---|---|
| ER | 6.3097 *** | 3.8034 *** | 6.3097 *** |
|  | (0.822) | (1.027) | (0.822) |
| GTI | 0.1669 *** | 0.1688 *** | 0.1669 *** |
|  | (0.050) | (0.038) | (0.050) |
| ISU | 0.0656 ** | −0.0152 | 0.0656 ** |
|  | (0.032) | (0.030) | (0.032) |
| FDI | 0.0013 *** | −0.0012 ** | 0.0013 *** |
|  | (0.000) | (0.001) | (0.000) |
| GFCF | 0.0312 *** | 0.0504 *** | 0.0312 *** |
|  | (0.009) | (0.010) | (0.009) |
| BDS | −0.5082 | 0.5233 | −0.5082 |
|  | (0.416) | (0.486) | (0.416) |
| _cons | 0.2733 *** | 0.2848 *** | 0.2733 *** |
|  | (0.050) | (0.055) | (0.050) |
| N | 90 | 90 | 90 |
| F |  | 96.2872 |  |
| R2_a | 0.8127 | 0.8643 | 0.8127 |
| N_g | 6.0000 | 6.0000 | 6.0000 |

Note: Standard errors in parentheses; ** $p < 0.05$, *** $p < 0.01$.

From the model fitting results, the goodness of fit was 0.8643, which achieved a high level of fit. In terms of variables, environmental regulation (ER) had a significant positive contribution effect on the level of new urbanization (NUR), where the coefficient was 3.8034 and was significant at a 1% level of significance, thus having a high degree of significance. This suggests that, on the one hand, according to the Porter hypothesis, environmental regulation increased the production cost of enterprises in a region in the short term, which lags economic growth, but the long-term panel data used in this paper confirms that over a longer time horizon, environmental regulation accelerated the green production efficiency of enterprises, thus increasing the revenue of enterprises in a region, which promoted regional economic development and thus accelerates the urbanization process of a region. On the other hand, environmental regulation can induce the transfer of industries in the region, and most enterprises tend to be located in urban areas. In the process of development and transfer, enterprises have to absorb talents and labor force as well as more advanced technological knowledge, which pulls the transfer of population and resources to urban areas, thus enhancing the new urbanization level of a region. Based on the above two aspects, in the process of urbanization, the government should focus on investment in environmental management and the development of eco-environmental technologies while transferring industries, thus effectively promoting the new urbanization development of a region in its ecological aspects.

In terms of the effects of the control variables, both the level of green innovation and fixed capital has a significant positive contribution to the level of new urbanization. The reason may be that green innovation is an important support to promote the construction of ecological civilization and promote high-quality development, while the city, as a gathering place of talents and enterprises, is destined to enhance green innovation without the introduction of talents and the development of enterprises, and therefore will become the main force to promote the development of new urbanization. In addition, the total amount of fixed capital formation indicates the economic foundation of a region. When a region is rich in fixed capital, the urbanization development will not lack the basic construction capacity to support it, and will inevitably have better development prospects. Foreign investment has a significant negative inhibitory effect on the level of new urbanization,

but the coefficient is only −0.0012, which has a very weak effect. The reason may be that the spillover effect of technology does not drive the rapid development of urbanization in local areas, but mainly relies more on the local innovation capacity and human capital level. Industrial upgrading and population basics, on the other hand, show insignificant effects on the development of new urbanization.

### 5.4. An Empirical Analysis of the Mediating Effect Regression Model of the Impact of Environmental Regulation on the Level of New Urbanization

To examine the mediating channel of regional low-carbon logistics capacity between environmental regulation and the level of new urbanization, firstly, according to model (1) and the analysis in the previous section, we found that environmental regulation has a significant contribution to the level of new urbanization. On this basis, model (2) and model (3) were constructed to test the adjustment effect of environmental regulation on regional low-carbon logistics capacity and the effect of regional low-carbon logistics capacity on the level of new urbanization, respectively. According to the reported results of model (2) and model (3) in Table 10, environmental regulation had a positive contribution to regional low-carbon logistics capacity with an estimated coefficient of 2.5858 and passed the 1% significance test. Next, regional low-carbon logistics capacity also had a significant positive driving effect on the level of new urbanization, with an estimated coefficient of 0.5467 and passed the 1% significance test. Finally, model (4) was constructed to include regional low-carbon logistics capacity in the mediating effect model to test the channels of action. In the last column of Table 10, the estimation results of model (4) are reported, and it was found that there was still a stable positive relationship between environmental regulation on the level of new urbanization compared to model (1), but the direct effect was 2.6114, which was significantly smaller than the total effect of model (1) of 3.8034, and the direct effect accounted for 68.66%, which laterally indicated the existence of some mediating effects, confirming the previous hypothesis Theory.

**Table 10.** Empirical analysis of regression model of intermediation effect.

| Variables | NUR(1) | LCLC(2) | NUR(3) | NUR(4) |
|---|---|---|---|---|
| ER | 3.8034 *** | 2.5858 *** | | 2.6114 *** |
| | (1.027) | (0.961) | | (0.975) |
| LCLC | | | 0.5467 *** | 0.4610 *** |
| | | | (0.109) | (0.110) |
| GTI | 0.1688 *** | 0.3726 *** | –0.0423 | –0.0030 |
| | (0.038) | (0.036) | (0.054) | (0.054) |
| ISU | –0.0152 | 0.0160 | –0.0090 | –0.0225 |
| | (0.030) | (0.028) | (0.028) | (0.027) |
| FDI | –0.0012 ** | –0.0001 | –0.0010 * | –0.0011 ** |
| | (0.001) | (0.001) | (0.001) | (0.000) |
| GFCF | 0.0504 *** | 0.0496 *** | 0.0338 *** | 0.0275 ** |
| | (0.010) | (0.009) | (0.011) | (0.011) |
| BDS | 0.5233 | 2.8335 *** | –1.3932 *** | –0.7829 |
| | (0.486) | (0.454) | (0.508) | (0.540) |
| _cons | 0.2848 *** | –0.3092 *** | 0.5031 *** | 0.4274 *** |
| | (0.055) | (0.051) | (0.055) | (0.060) |
| N | 90 | 90 | 90 | 90 |
| F | 96.2872 | 416.0867 | 109.8241 | 102.6058 |
| r2_a | 0.8643 | 0.9654 | 0.8792 | 0.8881 |
| N_g | 6.0000 | 6.0000 | 6.0000 | 6.0000 |

Note: Standard errors in parentheses; * $p < 0.1$, ** $p < 0.05$, *** $p < 0.01$.

According to the results of the analysis, a significant part of the positive contribution of environmental regulation to the level of new urbanization was transmitted through the regional low-carbon logistics capacity as a mediating mechanism. The explanation for this mediating effect is as follows: For one, environmental regulation, in the long run, can promote total factor productivity improvement in the logistics industry and improve technological progress and technical efficiency. The new development concept and the implementation of the national environmental protection policy and the strengthening of publicity will make consumers more favorable to green products and services, so the services and products with large carbon emissions will gradually exit the market; the logistics industry must make green improvements to meet the needs of today's market, and ultimately achieve a win-win harmony between environmental protection and industry development. Secondly, the improvement of regional low-carbon logistics capacity is closely related to the degree of specialization and concentration of the logistics industry in a region, because the division of labor specialization enables enterprises to concentrate on operating their core business, which can reduce the operating costs of logistics enterprises and at the same time improve the efficiency of transportation in a region, providing convenience for more efficient production by enterprises and the daily life of residents, so it also further improves the quality of life in towns and cities, and thus greatly improves the level of new urbanization. Finally, in a comprehensive view, environmental regulation has a strong intervention effect on the economic structure and resource allocation of a region, and can effectively control the carbon emission intensity of related industries, while the logistics industry, as a composite service industry integrating transportation, warehousing, logistics distribution, logistics packaging, and the information industry, are both important parts of the national economy and industry with high carbon emission intensity. Therefore, the logistics industry covers a wide range of areas and absorbs a large number of employed people, which plays a great role in promoting industrial structure upgrading, high-quality economic development, and new urbanization construction. Environmental regulation will also promote the healthy and smooth operation of the regional economy by controlling the logistics industry, changing the crude logistics operation mode, and forming a mutually coordinated development mode of low-carbon economic development and low-carbon logistics, which will also provide great convenience to the manufacturing, distribution and consumption fields in urban areas, and help solve the problems of urban transportation and comprehensive urban planning, providing security and support for urbanization construction. It is through such an intermediary transmission mechanism that environmental regulation can not only directly promote the new urbanization level of a region, but also indirectly influence the development of low-carbon logistics, thus redeploying the functions of cities and towns, optimizing the social environment of cities and towns, and realizing new urbanization development.

### 5.5. Robustness Tests

To ensure the robustness of the results, the explanatory variables were replaced in this paper, and the share of the urban population in the total population (%) was used to measure the level of new urbanization; the regression results are shown in Table 11. Comparing the data in Table 11 with those in Tables 9 and 10, we found that the results of the model were consistent, and the percentage of the direct effect of environmental regulation in Table 11 was 65.28%, which was almost consistent with the previous findings, thus indicating that the results of this paper were robust. Regional low-carbon logistics capacity as a mediating variable was an important channel through which environmental regulation in a region affected the level of new urbanization.

**Table 11.** Robustness test of the intermediation effect.

| Variables | NUR(1) | LCLC(2) | NUR(3) | NUR(4) |
|---|---|---|---|---|
| ER | 1.3622 *** | 2.5858 *** | | 0.8892 * |
| | (0.474) | (0.961) | | (0.463) |
| LCLC | | | 0.2121 *** | 0.1829 *** |
| | | | (0.051) | (0.052) |
| GTI | 0.0634 *** | 0.3726 *** | –0.0182 | –0.0048 |
| | (0.018) | (0.036) | (0.025) | (0.025) |
| ISU | 0.0441 *** | 0.0160 | 0.0457 *** | 0.0411 *** |
| | (0.014) | (0.028) | (0.013) | (0.013) |
| FDI | 0.0002 | –0.0001 | 0.0002 | 0.0002 |
| | (0.000) | (0.001) | (0.000) | (0.000) |
| GFCF | 0.0499 *** | 0.0496 *** | 0.0430 *** | 0.0408 *** |
| | (0.005) | (0.009) | (0.005) | (0.005) |
| BDS | 0.1165 | 2.8335 *** | –0.6096 ** | –0.4018 |
| | (0.224) | (0.454) | (0.236) | (0.256) |
| _cons | 0.2967 *** | –0.3092 *** | 0.3791 *** | 0.3532 *** |
| | (0.025) | (0.051) | (0.026) | (0.029) |
| N | 90 | 90 | 90 | 90 |
| F | 271.1649 | 416.0867 | 301.4124 | 267.7654 |
| r2_a | 0.9478 | 0.9654 | 0.9528 | 0.9544 |
| N_g | 6.0000 | 6.0000 | 6.0000 | 6.0000 |

Note: Standard errors in parentheses; * $p < 0.1$, ** $p < 0.05$, *** $p < 0.01$.

## 6. Conclusions and Policy Implications

### 6.1. Conclusions

This paper took the inter-provincial panel data of six central provinces from 2005 to 2019 as research samples, measured environmental regulation, regional low-carbon logistics capacity, and new urbanization levels in six central provinces, constructed an individual fixed-effects econometric regression model, studied and analyzed the impact of environmental regulation on the new urbanization level, and explored the principle of the mediated transmission mechanism of regional low-carbon logistics capacity between them. Our study led us to the following conclusions:

First, environmental regulation intensity and the level of new urbanization showed an overall upward trend during the period we studied; and there was a significant positive correlation between environmental regulation intensity and the level of new urbanization. Second, in terms of new urbanization development, the overall level of new urbanization in the six provinces we studied has shown a steady increase. Specifically for each province, Jiangxi Province had been at a high level of new urbanization because, in recent years, Jiangxi Province had highlighted the core concept of "people-oriented urbanization" and paid more attention to improving the quality and effectiveness of the citizenship of the transferred agricultural population, and the government had deepened the reform of the household registration system and further relaxed the conditions for key groups of people to settle in cities and towns, thus promoting the development of new urbanization in Jiangxi Province. The levels of new urbanization in Shanxi and Anhui provinces were also high because, in recent years, Shanxi Province had initially formed a pattern of an urban system with urban agglomerations as the main form and the coordinated development of large, medium, and small cities and towns. Since Anhui Province's integration into the Yangtze River Delta economic belt, its comprehensive strength had developed rapidly, and prefecture-level cities other than the provincial capital had paid more attention to the planning and expansion of urban layout to promote the development of new urbanization in the region. The new urbanization levels in Hubei, Hunan, and Henan provinces were

relatively low, while Henan province had the lowest value of this indicator among the six provinces studied. We all knew that Hubei Province and Hunan Province were two provinces with strong economic strength in central China, and their geographical locations were also transportation hubs in China, but in recent years, the urban population in these two provinces has tended to be saturated, and urbanization had encountered bottlenecks, especially Wuhan City in Hubei Province, which had recently been affected by a new coronavirus epidemic; there was a trend of population outflow, and the unbalanced development of various regions in the province had also led to the development of new urbanization being hampered. Henan Province had always been a province with a large ratio of rural population to the total population. Although the overall economic development had been rising rapidly in recent years, there had been a serious spillover of talents and the number of settled population had been growing slowly because of education pressure and other reasons. We suggest that Henan Province could open up new paths in the field of innovation, enhanced regional competitiveness, and became the backbone of the central provinces. Third, part of the impact of environmental regulation on the level of new urbanization was achieved by improving the low-carbon logistics capacity in the region, and it was here that regional low-carbon logistics capacity played a mediating role in transmission, which was also an important driver of urbanization development. Environmental regulation could significantly improve the low-carbon logistics capacity of a region through policy instruments and improved green production efficiency, indirectly contributing to the development of new urbanization in the region. As in recent years, various initiatives to expand rail transportation and waterway transportation, as well as the attempted transportation of hydrogen energy vehicles, have accelerated the speed of materials between urban and rural areas in an energy-efficient manner, thus driving the development of new urbanization in the entire region.

### 6.2. Policy Implications

6.2.1. Focus on the Construction of Ecological Civilization, and Improve the Environmental Regulatory Management System

The degree of ecological environment of a region can directly affect the attractiveness of an urban development and radiation level; the six central provinces in recent years in the rapid rise of urban development, in this stage, should implement relevant environmental policies, increase disciplinary efforts, and improve the cost of violations. Enterprises should pay more attention to low-carbon and environmentally friendly production methods, reduce carbon emissions at the source, and build a more livable urban living environment.

6.2.2. Emphasis on the Development of the Logistics Industry, to Enhance the Level of Supporting Infrastructure Construction

The logistics industry is now in a period of rapid development, but we still cannot ignore the high-quality development of the industry, especially since the level of logistics services still needs to be improved. As the key hub of Chinese transportation, the six central provinces are the center of national logistics distribution and transportation, so the rapid and high-quality development of the logistics industry in the region can have a huge pulling effect on the development of the national logistics industry. Specific initiatives are to introduce more cutting-edge logistics industry technologies from overseas in each region and improve the efficiency of energy utilization, while the central region should absorb the advantageous resources of the logistics industry in the surrounding Yangtze River Delta region and the Pearl River Delta region to promote the flow of employed people and the construction of transportation infrastructure facilities, which also accelerates the development of new urbanization.

6.2.3. Maintain the Advantages of New Urbanization Development and Drive the Coordinated Development of the Regional Green Economy

The six central provinces are the regions with the largest share of the population in China and also have a huge number of population movements between urban and rural

areas every year. The construction of new urbanization is related to the quality of life of the people and the long-term development of the region. First, Henan, Hubei, and Hunan can take advantage of local population resources to expand the scale of advantageous industries such as the logistics industry and promote the flow of people to urban areas; second, Anhui and Jiangxi can learn from the policy arrangements of neighboring developed provinces and introduce advanced production technologies in the logistics industry to enable local enterprises to implement low-carbon development paths; finally, Shanxi needs to accelerate the transformation of the local industrial structure, strengthen the policy binding on high carbon emission enterprises and expand the scale of urban areas to achieve balanced development. Combined with the development path with Chinese characteristics, it is necessary to take advantage of the diffusion role of the central provinces, strengthen the cooperation and exchange between regions, and cultivate and innovate new growth peaks in the urban economy, thus pulling together the coordinated development of the logistics industry and new urbanization in the surrounding areas.

**Author Contributions:** Methodology, Y.W.; software, Y.W., Y.H.; writing—original draft preparation, Y.W., Y.H.; writing—review and editing, Y.W., Z.Y.; formal analysis, Z.Y.; funding acquisition, Z.Y.; data curation, Y.W., Y.H. All authors have read and agreed to the published version of the manuscript.

**Funding:** This research received no external funding.

**Institutional Review Board Statement:** Not applicable.

**Informed Consent Statement:** Not applicable.

**Data Availability Statement:** The data presented in this study are openly available in China Statistical Yearbook, China Energy Statistical Yearbook, China Environmental Statistical Yearbook and Wind database.

**Conflicts of Interest:** The authors declare no conflict of interest.

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
