# Peer review of "The Effects of Environmental Regulation and Low-Carbon Logistics Capacity on the Level of New Urbanization in Six Central Provinces of China"

_sustainability, doi:10.3390/su141912686_

Round 1
Reviewer 1 Report
This is a timing and very interesting article, discussing the relationship between green low carbon logistics capacity and new urbanization taking a case study of six cities in the center of China. However, the article has several shortcomings. First, generally, if the article only focuses on the national situation of China, it more fits to publish it in a Chinese journal, while not in a renowned international journal like sustainability. The introduction part use almost two pages to describe the urbanization in China. However, usually the introduction should summarize the research gap in the international academics in the field of urbanization and logistic system development. Secondly, the logical development of the article should be improved. In the introduction part, the authors mentioned several questions at Lines 106-114. However, the conclusion part did not answer these questions clearly. Thirdly, some sentences are not naturally expressed in English language. I recommend not translating from a local language to English directly to improve readership. Fourthly, the introduction should be rewritten. This part is also important to describe what is the research gap in the relevant field by citing literature in the other countries. However, the first few paragraphs have almost no citiations. Please make it concise and only relevant to your research topic.
Just a few minor suggestions:
1. Line 602~603 the sentence said "previous studies" , please add the citation of previous studies.
2. Line 660~661 This sentence is like a general description, which should not be placed in the conclusion.
Reviewer 2 Report
Reviewer comments. The topic is interesting, and the author has done a great job in realizing the subject. However, there are few areas on the paper that is still lagging and should be addressed properly.
1 Abstract; the authors should motivate the why they chose of variables and Key words should match with the title.
2. Introduction section: The introduction presents several aspects that contextualize the research topic and presents the contribution of this work in comparison to previously published publications. The objective of the paper presented need more clarifications to suit reader to understand the main idea of the paper especially for the study
3. Literature Section: The authors highlighted the relevant theories that supports the research work and provided evidence in the literature that support the adopted theories. I however, suggest that the literature should be put under subheadings to clearly highlight the different aspects of the study. Again, there is need for more recent studies ranging from 2018-2022 to motivate the study properly. The entire study is too scanty and the related literature is not exhausted
Ø
Mitigating emissions in India: accounting for the role of real income, renewable energy consumption and investment in energy. 670216917.
Ø Determinants of CO2 emissions in the BRICS economies: The role of partnerships investment in energy and economic complexity. Sustainable Energy Technologies and Assessments, 51, 101907.
Ø Environmental consequences of foreign direct investment influx and conventional energy consumption: evidence from dynamic ARDL simulation for Turkey. Environmental Science and Pollution Research, 1-14.
Ø Accounting for the combined impacts of natural resources rent, income level, and energy consumption on environmental quality of G7 economies: a panel quantile regression approach. Environmental Science and Pollution Research, 29(2), 2806-2818.
Ø Sterling insights into natural resources intensification, ageing population and globalization on environmental status in Mediterranean countries. Energy & Environment, 0958305X221083240.
Ø Consumption-based carbon emission and foreign direct investment in oil-producing Sub-Sahara African countries: the role of natural resources and urbanization. Environmental Science and Pollution Research, 29(9), 13154-13166.
Methodology
1. The variables used in the model should be justified
2. More benefit of the various techniques utilized should be stated. And if possible, their equations should be added to the revised manuscript to enrich the quality.
Discussion
1. The discussion is well written, but the authors should like their findings to the previous studies in the literature.
The policy implication and conclusion section:
1. The authors are advised to provide practical policy to help the esteemed reader.Reviewer comments. The topic is interesting, and the author has done a great job in realizing the subject. However, there are few areas on the paper that is still lagging and should be addressed properly.
1 Abstract; the authors should motivate the why they chose of variables and Key words should match with the title.
2. Introduction section: The introduction presents several aspects that contextualize the research topic and presents the contribution of this work in comparison to previously published publications. The objective of the paper presented need more clarifications to suit reader to understand the main idea of the paper especially for the study
3. Literature Section: The authors highlighted the relevant theories that supports the research work and provided evidence in the literature that support the adopted theories. I however, suggest that the literature should be put under subheadings to clearly highlight the different aspects of the study. Again, there is need for more recent studies ranging from 2018-2022 to motivate the study properly. The entire study is too scanty and the related literature is not exhausted
Ø
Mitigating emissions in India: accounting for the role of real income, renewable energy consumption and investment in energy. 670216917.
Ø Determinants of CO2 emissions in the BRICS economies: The role of partnerships investment in energy and economic complexity. Sustainable Energy Technologies and Assessments, 51, 101907.
Ø Environmental consequences of foreign direct investment influx and conventional energy consumption: evidence from dynamic ARDL simulation for Turkey. Environmental Science and Pollution Research, 1-14.
Ø Accounting for the combined impacts of natural resources rent, income level, and energy consumption on environmental quality of G7 economies: a panel quantile regression approach. Environmental Science and Pollution Research, 29(2), 2806-2818.
Ø Sterling insights into natural resources intensification, ageing population and globalization on environmental status in Mediterranean countries. Energy & Environment, 0958305X221083240.
Ø Consumption-based carbon emission and foreign direct investment in oil-producing Sub-Sahara African countries: the role of natural resources and urbanization. Environmental Science and Pollution Research, 29(9), 13154-13166.
Methodology
1. The variables used in the model should be justified
2. More benefit of the various techniques utilized should be stated. And if possible, their equations should be added to the revised manuscript to enrich the quality.
Discussion
1. The discussion is well written, but the authors should like their findings to the previous studies in the literature.
The policy implication and conclusion section:
1. The authors are advised to provide practical policy to help the esteemed reader.
Reviewer 3 Report
The topic is quite interesting and worthwhile studying, and some parts are needed revision before it can achieve the level of publish.
1. 1 The paper is quite lengthy. It can be concise especially in introduction and LR, where too many words are not very pertinent or interesting.
2. 2 Theoretical discourse and contribution are not very clear. Although starting from Porter hypothesis, it is unclear how the paper contributes to theoretical discourse. Just prove the hypothesis or develop something new? On the other, lots of discussion is irrelevant, for example, about FDI, spillovers, etc.
3. 3 It needs careful thinking about indicators especially in ecology dimension, as gross value may not be good.
4. 4 There is a need of statement why authors choose middle regions, and the title should be adjusted accordingly to make it more transparent.
5. 5 There is a need for heavy editing,
- some expressions are unclear, e.g., “in terms of new urbanization development, the whole six provinces are steadily increasing”.
- Some format is not English use, e.g., 《》
Round 2
Reviewer 3 Report
Thank you for giving me a chance to review the paper. I have read through it and the revision is not complete, which requires minor revision.
1. The paper does not answer the previous question about how it makes marginal theoretical contributions.
2. What is the low carbon logistics capacity? Better to define and explain that.
Author Response
请参阅附件。
